# A Cross-Sectional Survey Study Examining the Provision of Continuous Glucose Monitoring Education in U.S. Doctor of Pharmacy Programs

**DOI:** 10.3390/pharmacy10060174

**Published:** 2022-12-16

**Authors:** Emily Knezevich, Kevin T. Fuji, Krysta Larson, Gabrielle Muniz

**Affiliations:** 1Department of Pharmacy Practice, Creighton University School of Pharmacy & Health Professions, Omaha, NE 68178, USA; 2Pharm.D. Candidates, Creighton University School of Pharmacy & Health Professions, Omaha, NE 68178, USA

**Keywords:** continuous glucose monitoring, diabetes, pharmacy education, pharmacy students, diabetes education, diabetes self-management

## Abstract

Continuous glucose monitoring (CGM) is used to help patients with diabetes and their healthcare providers more effectively manage care. CGM use is expanding to all healthcare settings where pharmacists practice and new pharmacy graduates may increasingly be asked to assist patients utilizing CGM devices and assess diabetes management through the interpretation of CGM data. The purpose of this study was to describe CGM education across Doctor of Pharmacy (Pharm.D.) programs in the United States. An online survey was administered to 139 accredited Pharm.D. programs. Information was solicited about CGM education, including curricular placement, course type, hands-on experience, and credential(s) of faculty providing the education. Fifty-seven programs responded with 51 (89.5%) providing CGM education for a median of 1.0 h. Of programs providing detailed responses, content was delivered in required (60.4%) or elective (45.8%) lectures as well as experiential settings (41.7%). Education occurred most frequently in the third year (58.3%), followed by the second (43.8%) and fourth (37.5%) years. Thirty-one (66.0%) programs were taught by a faculty member with an advanced diabetes credential. The results from this study confirm that there is an ongoing need to examine optimal amount, timing, and methods for providing CGM education.

## 1. Introduction

Diabetes continues to be a significant health problem in the United States (US), impacting over 37 million people with an estimated total cost (direct and indirect) of USD 327 billion [1]. Uncontrolled diabetes increases the risk of cardiovascular disease, nephropathy, neuropathy, retinopathy, and stroke [2]. However, the effective management of diabetes by patients and their healthcare providers can help avoid both short- and long-term complications of the disease.

A relatively recent innovation in diabetes care is the use of continuous glucose monitoring (CGM), which provides real-time tracking and recording of blood glucose changes that allow patients and their diabetes care providers to optimize self-management and treatment decisions based on more comprehensive data than the traditional self-monitoring of blood glucose [3,4]. Studies have demonstrated that utilizing CGM results in improved clinical outcomes including reductions in hemoglobin A1c and hypoglycemic events, and improvement in the time in therapeutic range for blood glucose [5,6,7]. These benefits have led the American Diabetes Association to recommend CGM use for patients with diabetes who require insulin therapy and to even consider use in patients with type 2 diabetes who do not require insulin therapy [8].

The benefit of CGM is contingent on the patient having appropriate diabetes education, training, and support. Pharmacists already play a role in diabetes education, and it is reasonable to expect this will expand to include assisting patients with CGM devices [9,10]. There are already positive impacts described from the emerging literature about pharmacist involvement with CGM both in pharmacist-led efforts and as part of an interprofessional care team [11,12,13,14,15]. While most of this literature is based in the ambulatory care setting where most patients with diabetes are cared for, the literature also describes the increasing use of CGM in the hospital setting, providing another opportunity for pharmacist engagement [16]. Additionally, pharmacists in the community setting may be educating and assisting patients with CGM device use as CGM devices are increasingly being covered by a patient’s prescription drug benefit in addition to the more traditional coverage through a patient’s durable medical equipment benefit. In fact, one CGM manufacturer has reported that approximately 50% of sales are occurring at the pharmacy [17].

It is clear that pharmacists across multiple care settings will need the knowledge and skills to assist patients in using CGM to manage their diabetes care. However, as an emerging technology with evolving use, it is unclear if CGM education is currently being provided in Doctor of Pharmacy (Pharm.D.) training. Although it is not uncommon for pharmacy education to lag behind changes in practice, this is a crucial practice trend that may require a more rapid integration into curricula to ensure that the needs of diabetes patients seen across all care settings are met. Additionally, the provision of education surrounding the pharmacist’s role in analyzing information collected by CGM and relating this to medication-related problems and patient-centered goals to develop a comprehensive care plan aligns with both the Pharmacist Patient Care Process and Core Entrustable Professional Activities that all pharmacy graduates should demonstrate [18,19]. The purpose of this study was to describe CGM education in US Pharm.D. programs.

## 2. Materials and Methods

A cross-sectional study design was used in which an online survey was distributed to all 139 accredited US Pharm.D. programs from December 2020 to February 2021. An initial invitation e-mail was sent to one faculty member at each Pharm.D. program. Faculty were identified using program websites with primary selection criteria based on individuals with either the CDCES or BC-ADM credential. If no faculty met this criterion, the secondary selection criteria included an indication of a diabetes research focus or teaching diabetes content within the program. It was believed this targeted approach would reach individuals who were either teaching this content directly or were knowledgeable about the content being taught. The invitation e-mail included instructions to complete the online questionnaire within the Qualtrics platform. It also advised the recipient to forward the e-mail to a more appropriate individual if needed. Two weeks after the initial e-mail invitation, a follow-up reminder e-mail was sent to non-responders, and two weeks after the reminder e-mail, a follow-up phone call was made to non-responders. All responses were confidential, but not anonymous. The study protocol was reviewed by the authors’ institutional review board and approved as “exempt”.

### 2.1. Survey Development

The survey items were self-developed to solicit information about CGM education within each program’s curriculum. The total number of survey questions differed upon a participant’s response to an initial question asking whether their program provided CGM education. Programs providing CGM education responded to five additional questions about number of hours provided (free-text response), curricular placement (P1, P2, P3, and/or P4 year), type of course(s) in which the information was provided (didactic vs. experiential; required vs. elective; lecture vs. experiential vs. lab), if students had the opportunity to have hands-on experience with a CGM device (yes or no), and if the individual(s) providing CGM education had an advanced diabetes credential (e.g., Certified Diabetes Care and Education Specialist (CDCES) or Board Certified in Advanced Diabetes Management (BC-ADM)). Programs that were not providing CGM education responded to two additional questions. The first asked if they planned to provide CGM education in the future (yes or no) and the second was a free-text response asking them to specify why they were not currently providing CGM education. 

The face validity of the survey was established as the primary author is a pharmacist with a CDCES, actively practicing in an endocrinology clinic, and recognized as an expert in the use of technology by pharmacists to support diabetes management. Additionally, the survey was pilot tested with a pharmacy faculty member who is a CDCES. After editing for clarity, the finalized survey was placed on the Qualtrics platform.

### 2.2. Data Analysis

The data were exported from Qualtrics to SPSS Statistics version 27 (IBM Corporation) for analysis. The data were primarily analyzed descriptively, reporting frequency counts and percentages for all categorical variables. Hours of CGM education was reported as a median and range. A sub-analysis was conducted to characterize curricular placement and type of course in which CGM education was provided. A post hoc analysis was conducted to compare the distribution of respondents to non-respondents relative to the program characteristics. A chi-square test was used with a significance level of *p* ≤ 0.05. Finally, the free-text responses describing the reasons for not currently offering CGM education were categorized and reported as frequency counts. 

## 3. Results

A total of 57 programs responded to the survey (41% response rate). The majority of the programs (n = 51, 89.5%) indicated providing CGM education. There were varying levels of response to each of the follow-up questions, and for these questions, the “n” is noted. CGM education was provided for a median of 1.0 h (n = 46) with a range of 0.1 to 30 h. 

Most respondents were in private programs (n = 33, 57.9%), had a 4-year program structure (n = 46, 80.7%), and were geographically distributed across the US. Table 1 displays the characteristics of the respondents in comparison to the population of pharmacy programs. The post hoc analysis revealed no significant differences in program characteristics between the survey respondents and non-respondents. 

### 3.1. Course Type and Curricular Placement of CGM Education (n = 48)

CGM education was provided most frequently in required lectures (n = 29, 60.4%), followed by elective lectures (n = 22, 45.8%), experiential settings (n = 20, 41.7%), and required skills laboratory courses (n = 6, 12.5%).

CGM education was provided by three programs in the first year (6.3%), 21 in the second year (43.8%), 28 in the third year (58.3%), and 18 in the fourth year (37.5%). Twenty programs indicated providing CGM education in multiple years: first and third year (n = 1, 2.1%), first and fourth year (n = 2, 4.2%), second and third year (n = 2, 4.2%), second and fourth year (n = 3, 6.3%), second, third, and fourth year (n = 2, 4.2%), and third and fourth year (n = 10, 20.8%). Additional detail regarding curricular placement and course type is provided in Table 2. 

### 3.2. Hands-on CGM Education (n = 48)

Hands-on education was provided in 16 programs (33.3%). Of the 16 programs providing hands-on CGM education, 10 provided it in a required lecture (62.5%), 10 in an elective lecture (62.5%), two in a required skills laboratory (12.5%), and eight in an experiential setting (50%). No program provided hands-on CGM education in the first year, seven provided it in the second year (43.8%), 12 provided it in the third year (75%), and eight provided it in the fourth year (50%). 

### 3.3. Expertise of Faculty Providing CGM Education (n = 47)

Thirty-one programs (66.0%) had their CGM education provided by a faculty member with an advanced diabetes credential. 

### 3.4. Programs Not Providing CGM Education

Of the six programs that indicated not providing CGM education, only one noted a plan to provide this education in the future. Four programs cited a lack of time or shared that they have not asked their faculty member(s) who possess an advanced diabetes credential to provide CGM education. One program did not include a reason for not providing CGM education.

## 4. Discussion

The vast majority (89.5%) of survey respondents indicated providing some form of CGM education in their Pharm.D. curriculum, although with a median of 1.0 h spent on the topic, there may be opportunities for growth. While there is no recognized optimal number of hours for this type of instruction, a recent paper describing a CGM educational module for students and pharmacists included a little over three hours of instruction along with associated home study [20]. This instruction included hands-on training, a typical component of diabetes education for pharmacy students [21,22]. Only a third of the survey respondents provided hands-on experience with CGM devices, suggesting that any increase in hours spent on CGM education should include hands-on training and potentially in skills laboratory courses (with only six respondents indicating CGM education in this type of course). The addition of this content into pharmacy school curricula must be balanced with the challenge of avoiding curricular hoarding [23]. The inclusion of basic concepts of CGM with a brief hands-on experience would allow all pharmacy students to have exposure to the technology, but many will achieve mastery understanding in postgraduate or on-the-job training. 

Multiple studies have described the benefit of the pharmacist-guided implementation of CGM. In a study that compared pharmacist utilization of CGM in patients with uncontrolled type 2 diabetes to the usual care provided by a physician, a demonstrated clinically and statistically significant improvement in hemoglobin A1c was noted [12]. As a profession, pharmacists are equipped to couple disease state education with pharmacotherapeutic recommendations to optimize the care of people with chronic diseases such as diabetes. In order for pharmacists to remain essential members of the interprofessional healthcare team, the profession must ensure students are kept up-to-date on current management trends. With the increase in published literature, practice changes, and health insurance coverage changes that suggest pharmacists are engaging with patients using CGM devices and will continue to do so in the future, there may be a need to shift CGM education from elective courses (45.8% of respondents) to required courses (60.4% of respondents). From a pedagogical standpoint, Pharm.D. programs may also benefit from providing CGM education across multiple years, as less than half (41.7%) of respondents indicated multi-year education efforts. 

Most respondents had their CGM education provided by an individual with an advanced diabetes credential. While this type of credential is not needed to teach this content, these individuals may have more direct experience with CGM devices. There may be benefit in supporting faculty to obtain an advanced diabetes credential or to consult practitioners with actual CGM device experience to support the design and delivery of this content.

### Study Limitations

There was a relatively low response rate to the survey (41% overall and slightly less for more detailed questions about the CGM education provided). This level of response is not uncommon, but limits the generalizability of the study findings. It is possible that non-responders chose not to respond due to a lack of CGM education being integrated into their curricula. If true, the study findings could overestimate the number of programs providing CGM education. Positively, there was a relatively diverse representation of programs that was comparable to the population of pharmacy programs in the US. 

The face validity and content validity of the survey could have been more strongly established through consultation with additional experts in the field. To this point, the survey did not solicit information about the specific CGM topics covered (e.g., technical use of the device, downloading and interpreting data, etc.). Future research must focus on not just the provision of CGM education, but the specific topics presented, the amount of time spent on each topic, and the effectiveness of different teaching modalities (lecture, hands-on, experiential) on student knowledge and confidence regarding CGM. 

Programs with varying program structures responded to the survey. While respondents could select the year(s) in which CGM education is provided, a second year offering in a three-year program may differ from a second year offering in a four-year program. Regardless, CGM education was generally provided after the first year of the curriculum. 

Self-reported data were collected, and some respondents may have provided inaccurate estimates of CGM education. While alternative approaches were considered, including sending the recruitment e-mail to individuals in charge of curriculum at each program, this information was not readily available through most program websites. Additionally, while these individuals may have an overall view of their curriculum, they may not be knowledgeable about the specific content addressed within each course. This limitation may have been partially mitigated by asking individuals to send the survey invitation on to a faculty member with the knowledge needed to respond accurately to the survey.

## 5. Conclusions

While most survey respondents indicated providing CGM education, the amount of time devoted to the topic within US Pharm.D. programs varied. With the growth of CGM use across care settings and pharmacists’ roles as members of the care team helping patients with diabetes to manage their care, it is reasonable to expect that many pharmacists will assist with the use of CGM devices. Future studies should focus on specific CGM topics and the optimal ways to present that content to students. 

## Figures and Tables

**Table 1 pharmacy-10-00174-t001:** Demographic characteristics of the study sample compared to the population of programs and non-respondents.

Characteristic	Study Sample, n (%)	Population of Programs, n (%)
**Program Type**		
Private	33 (57.9)	68 (49.3)
Public	24 (42.1)	70 (50.7)
**Program Structure**		
3-year program	7 (12.3)	17 (12.3)
4-year program	46 (80.7)	114 (82.6)
6–7-year program	4 (7.0)	7 (5.0)
**Geographic Location**		
Northeast	12 (21.1)	26 (19.1)
Midwest	14 (24.6)	31 (22.8)
South	18 (31.6)	53 (39.0)
West	13 (22.8)	26 (19.1)

**Table 2 pharmacy-10-00174-t002:** CGM education by year and type of course.

	P1,	P2,	P3,	P4,	Total
	n (%, Year)	n (%, Year)	n (%, Year)	n (%, Year)	
Required lecture, n (% course type)	3 (10.3)	12 (41.4)	14 (48.3)	0 (0)	29
	(100)	(54.5)	(41.2)	(0)	
Required skills lab, n (% course type)	0 (0)	5 (83.3)	1 (16.7)	0 (0)	6
	(0)	(22.7)	(2.9)	(0)	
Elective lecture, n (% course type)	0 (0)	4 (19.0)	17 (81.0)	0 (0)	21
	(0)	(18.2)	(50.0)	(0)	
Experiential, n (% course type)	0 (0)	1 (4.8)	2 (9.5)	18 (85.7)	21
	(0)	(4.5)	(5.9)	(100)	
Total	3	22	34	18	77

## Data Availability

The data presented in this study are available on request from the corresponding author. The data are not publicly available due to use for ongoing analyses that have not yet been published.

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
