# Peer review of "A Cross-Sectional Survey Study Examining the Provision of Continuous Glucose Monitoring Education in U.S. Doctor of Pharmacy Programs"

_pharmacy, 2022, doi:10.3390/pharmacy10060174_

Round 1

Reviewer 1 Report (New Reviewer)

Thank you for the opportunity to review this manuscript. Given the expanding role of pharmacists in ambulatory care settings as well as provider status initiatives that allow us to bill for clinical services, this manuscript is a timely addition to the literature as well as information for pharmacy education. 

Overall, the manuscript is well-written. I offer some comments to enhance the clarity and information in the manuscript.

In the introduction, consider aligning CGM not only with the core EPAs but also with the Pharmacists’ Patient Care Process (PPCP), since it is a required element vs the EPAs (not yet required).

Methods:

  1. Please add in the total number of items on the survey as well as any scaling of items or if they were all simply categorical in nature.
  2. Please add an overview of your data analysis in the methods section (instead of interspersing it throughout the results). Include the software utilized for analysis and planned/implemented analyses.

Results:

  1. In Table 2, consider presenting your data not only as “n” but also include the % to aid interpretation.
  2. Did you ask any questions about perceptions of integrating CGM to respondents? If not, that is fine - that may be a future consideration as integration often relates to perceived importance / necessity to practice.

Discussion: 

  1. In the discussion, you allude to the limited time spent on CGM as well as the unknown variable of the non-respondent programs. Given the recent discussion across the academy on the ever expanding problem of curricular hoarding / stuffing, etc., how much is reasonable? How do we adequately address practice changes while still preparing an entry-level pharmacist? Can the authors posit a starting amount or areas of content that could be considered for an entry level pharmacist, with the understanding that future jobs or postgraduate education may build upon this?
  2. There is limited literature presented related to CGM (two studies cited in the discussion). Are there other elements (perhaps to address my first comment?) that you could utilize supporting literature to enhance the contextualization and findings? Also, you could consider addressing the need for credentialed faculty to teach specific content areas using literature. Or, the problem of curricular expansion and determining what to include vs not?

Author Response

Dear Reviewer, 

Thank you for your very helpful feedback in improving our manuscript.  Please see the following comments addressing how each comment was incorporated into the paper.  

reviewer comment 1. In the introduction, consider aligning CGM not only with the core EPAs but also with the Pharmacists’ Patient Care Process (PPCP), since it is a required element vs the EPAs (not yet required).

Author response: Added information about alignment of CGM with the PPCP.

Reviewer comment 2: Please add in the total number of items on the survey as well as any scaling of items or if they were all simply categorical in nature.

Author response: Additional detail regarding the survey is now provided in the Methods section and a sub-section has been created to highlight this information.

Reviewer comment 3: Please add an overview of your data analysis in the methods section (instead of interspersing it throughout the results). Include the software utilized for analysis and planned/implemented analyses.

Author response: The analyses were moved to all be included in the Methods section. The software used is already noted in the Methods section – SPSS Statistics version 27.

Reviewer comment 4: In Table 2, consider presenting your data not only as “n” but also include the % to aid interpretation.

Author response: As there are two different sets of percentages to include (both year and course type), percentage for course type is now included to the right of the “n” and percentage for the year is now included under the “n”.

Reviewer comment 5: Did you ask any questions about perceptions of integrating CGM to respondents? If not, that is fine - that may be a future consideration as integration often relates to perceived importance / necessity to practice.

Author response: This was not asked in the survey.

Reviewer comment 6: In the discussion, you allude to the limited time spent on CGM as well as the unknown variable of the non-respondent programs. Given the recent discussion across the academy on the ever expanding problem of curricular hoarding / stuffing, etc., how much is reasonable? How do we adequately address practice changes while still preparing an entry-level pharmacist? Can the authors posit a starting amount or areas of content that could be considered for an entry level pharmacist, with the understanding that future jobs or postgraduate education may build upon this?

Author response: A statement has been added to the Discussion regarding the need for inclusion without resultant curricular stuffing. Additional studies have also been included to discuss the pharmacist’s role and what would be needed as a basic understanding for pharmacists to successfully work with patients who have CGM.

Reviewer comment 7: There is limited literature presented related to CGM (two studies cited in the discussion). Are there other elements (perhaps to address my first comment?) that you could utilize supporting literature to enhance the contextualization and findings?

Author response: Added in one additional study to stress positive impact of pharmacists in working with CGM.  

Reviewer comment 8: Also, you could consider addressing the need for credentialed faculty to teach specific content areas using literature. Or, the problem of curricular expansion and determining what to include vs not?

Author response: This has been addressed with revisions to the Discussion regarding being aware of curricular hoarding while still including basic understanding and hands-on experience.

Again, we appreciate the suggestions you have made and feel they have enhanced our work greatly. 

Reviewer 2 Report (New Reviewer)

Author Response

Dear Reviewer,

Thank you for the time and effort you provided to review our manuscript.  We appreciate you comments and believe we have addressed them thoroughly in our responses below. 

Reviewer comment 1: Overall the introduction was well written. For context, do you want to describe the amount of diabetes education that pharmacy schools provide? For example: Mansell K, Mansell H, Neubeker W, Drake H. Students' perceptions of and amount of diabetes education in Canadian schools of pharmacy. Curr Pharm Teach Learn. 2017 May;9(3):376-382. (Canada) You may want to see if there is a more recent article highlighting this, and also look for USA data.

Author response: Since this data looked at US pharmacy schools, we do not feel that the Canadian data is pertinent.  No similar data from US pharmacy schools has been published.

Reviewer comment 2: With respect to the methods section, there are a few questions that need to be answered. How were 139 schools chosen? Is that the total amount of accredited pharmacy schools in the USA?

Author response: The word “all” was added to demonstrate this survey was to all accredited programs.

Reviewer comment 3: Is having just one pharmacist (CDCES) enough for face validity? And pilot testing with only 1 other pharmacy faculty member? I would hazard to guess that the questionnaire could have been more informative had it been built and tested with more input. I do not believe you can state you did face validity when in reality you did not ask anyone for input – just yourself. Hence, I would make mention in the limitations section that multiple sources were not consulted for face validity and content validity. Unfortunately, this limits the utility of your developed tool.

Author response: Given that the primary author is an expert in this area who recently co-authored a publication with a team of experts nationally on the use of technology by pharmacists to support diabetes care, the authors feel that this (in addition to the piloting by another pharmacist with the CDCES credential) should be sufficient to establish face validity, which is a qualitative determination. This information has been added to the Methods. However, the authors also acknowledge that multiple sources have value in providing a more comprehensive review of a survey and have added a short statement in the Limitations as it pertains to stronger establishment of face and content validity.

Reviewer comment 4: Also, did you look to see if there was already a questionnaire out there you could use? If so, and one did not exist, you should mention that.

Author response: It was previously noted that this is the first such survey and therefore required self-development.

Reviewer comment 5: With respect to the survey, was it paper based or web based? If web-based, what platform was used?

Author response: The survey being web-based and hosted on the Qualtrics platform was previously included.

Reviewer comment 6: Were responses anonymous?. If so, how did you maintain anonymity? What format were the questions in? Were the Likert questions? Were there any open ended questions? How many questions in total were asked? How did you choose this amount? Please provide a more detailed description of the survey in the methods section.

Author response: The survey was confidential but not anonymous.  Additional detail regarding the survey design have been added to a specific sub-section in the Methods.

Reviewer comment 7: I think you should remove the p value from table 1 as it is kind of irrelevant.

Author response: P-values have been removed.

Reviewer comment 8: I think the first paragraph should state right away how many schools say they provide CGM education, and how many did not. From your text: “The majority of programs (n=51, 89.5%) indicated providing CGM education for a 114 median of 1.0 hours (n=46) with a range of 0.1 to 30 hours”. I am confused with the difference between 46 and 51. Does this mean that 51 schools said yes, but only 46 people answered this question?

Author response: The first and second paragraphs have essentially been switched per the reviewer’s request.  A clarifying sentence has been added regarding the varying response rates for each survey question.

Reviewer comment 9: Also, how was this question asked? Was it free form (open ended) or MCQ?

Author response: This was a free response question.  This is noted in the updated Methods section.

Reviewer comment 10: I am curious as to how the 1 hour was arrived at? Also, if the range is 0.1 to 30, and the median is the middle value, how could the median be 1? I am confused by this – could this be made into a table? I am not sure, it is just an idea.

Author response: The median is 1 because it was the middle value of the 46 responses received for that survey item.  The range indicates that the lowest number of hours of CGM education provided was 0.1 and the highest was 30.  The authors feel that noting the median is a fairly standard was of reporting data, particularly for those with extreme ranges that could skew the reporting of a mean, and that a table would not be beneficial.   

Reviewer comment 11: From your manuscript: “The vast majority (89.5%) of…” It is always a good idea to put an n in brackets behind this (for example, n=51/57)

Author response: The authors appreciate this suggestion but believe that given the clarification regarding the varying response rates to specific questions and also considering the journal formatting, that no change is necessary.

Reviewer comment 12: You should clarify by what you mean with respect to CGM – real-time or intermittently scanned? And did you delineate between these 2 (or define the differences) in your questionnaire?

Author response: There was not a differentiation made between the two in the survey.  The authors do not feel this would change responses and the overall goal of the survey was to identify if CGM content was being covered at all. Added the words “any form of” to line 191 to identify that no differentiation about type was made. 

Reviewer comment 13: I am not sure if the conclusion should be that the time spent teaching is minimal. This Is based purely on conjecture. Stick to the facts that you found out – most schools surveyed offer some education and to varying degrees. Further study is needed to answer some of the questions that you have posed yourself, but your questionnaire doesn’t answer the question of if it is enough time spent or .

Author response: The authors appreciate the reviewer’s comments and have changed “minimal” to “varied”.

Reviewer comment 14: As an aside, in the copy I reviewed, there were multiple places where some text was highlighted in yellow. I am not sure why that was.

Author response: Instructions for resubmission requested changes be highlighted.  Highlighting has been removed

Round 2

Reviewer 2 Report (New Reviewer)

I believe that most of the comments I suggested were addressed. I do feel that this small study has some significant methodological concerns , but overall the information gleaned could be useful for other schools of pharmacy to know about. 

On the other hand, the response rate is low, and it was only a simple 4 or 6 question questionnaire, and so methodologically it is of low value. I feel the authors could do a better job of highlighting the limitations to the study (face validity, questionnaire design in particular) and downplaying the results and conclusions. For instance, there is no mention in the conclusions that to answer some of the questions they asked, that more study is required.

Author Response

Dear Reviewer, see below for author response to your review: 

Reviewer: I believe that most of the comments I suggested were addressed. I do feel that this small study has some significant methodological concerns, but overall the information gleaned could be useful for other schools of pharmacy to know about.  On the other hand, the response rate is low, and it was only a simple 4 or 6 question questionnaire, and so methodologically it is of low value. I feel the authors could do a better job of highlighting the limitations to the study (face validity, questionnaire design in particular) and downplaying the results and conclusions. For instance, there is no mention in the conclusions that to answer some of the questions they asked, that more study is required.

Author response: The authors appreciate the reviewer’s additional comments and agree that while there are limitations to the study, it provides useful information to help other schools of pharmacy in their decision-making regarding introducing CGM content into their curricula.  Respectfully, the authors feel that the revisions that have been made have softened the results so that it does not over-reach with the purely descriptive data that was collected, and utilizes existing literature to place the results in an appropriate context.  The Limitations sub-section has been revised to remove or revise sentences that attempt to mitigate limitations in a way that may overly soften the weaknesses of the survey.  The Conclusion has also been edited to finish by noting the need for future research to explore the offering of CGM content in more depth than was solicited in the current survey.    

This manuscript is a resubmission of an earlier submission. The following is a list of the peer review reports and author responses from that submission.

Round 1

Reviewer 1 Report

The authors provide results of a survey to Pharm D programs on the delivery of CGM education.  The methods and results are clearly stated, and the discussion provides context.  No revisions are suggested.

Reviewer 2 Report

The manuscript is well-written, however, the authors do not provide information that is not already known and published by other research groups where information about the education program is provided and not only a yes/no CGM education as in this manuscript. The authors could provide more information about the topics in the education program.

Reviewer 3 Report

Title:

Appropriate

Abstract:

Appropriate

Introduction: 

I would recommend including something related to CAPE/EPA that describe how pharmacists are already involved in device education and management and how this satisfies outcomes related to curricula - better use case for SOP/COPs to start including.

Methods:

Curious as to why the CDCES or BC-ADM was chosen as the representative to send it to, as opposed to a curriculum committee chair, an assessment person, maybe even a skills lab person, etc. I'm assuming the authors were hoping to increase responses by sending it to the person most likely to care about CGMs. Might be valuable to address here in the methods.

Results:

Lines 131-141 are challenging to read - can you put this in a chart or something to make it a bit more easily digestible? 

Discussion:

I wonder why the authors have not expounded upon their recommendations for how this should be included in a curriculum, such as a need for hands-on education on CGMs whether in skills/simulation and a reinforcement in experiential. I'd like to see a more concrete "this is what PharmD programs should do with this information" outlined.

Conclusion:

Okay

Tables & Figures:

Okay - I like that you compared your population to PharmD programs.